# Growth Enhancement and Resistance of Banana Plants to *Fusarium* Wilt Disease as Affected by Silicate Compounds and Application Frequency

**DOI:** 10.3390/plants13040542

**Published:** 2024-02-16

**Authors:** Md Aiman Takrim Zakaria, Siti Zaharah Sakimin, Mohd Razi Ismail, Khairulmazmi Ahmad, Susilawati Kasim

**Affiliations:** 1Department of Crop Science, Faculty of Agriculture, Universiti Putra Malaysia, Serdang 43400, Malaysia; aimanupm_91@yahoo.com (M.A.T.Z.); razi@upm.edu.my (M.R.I.); 2Institute of Tropical Agriculture and Food Security (ITAFoS), Universiti Putra Malaysia, Serdang 43400, Malaysia; 3Department of Plant Protection, Faculty of Agriculture, Universiti Putra Malaysia, Serdang 43400, Malaysia; khairulmazmi@upm.edu.my; 4Department of Land Management, Faculty of Agriculture, Universiti Putra Malaysia, Serdang 43400, Malaysia; susilawati@upm.edu.my

**Keywords:** *Fusarium*, banana, silicate, proline, growth

## Abstract

The amendment of soils with silicate (Si) compounds is essential to promote growth performance and control *Fusarium* wilt disease in bananas. Two successive greenhouse trials were conducted at the experimental farm of the University of Putra Malaysia. The treatments were arranged in split plots using a randomized complete block design (RCBD) with four replicates to investigate the effects of Si compounds and application frequency on controlling FOC. Si compounds were used at a constant concentration of 0.1%: T0 (control), T1 (13% SiO_2_:20% K_2_O), T2 (26.6% SiO_2_:13.4% K_2_O) and T3 (36.2% SiO_2_:17% Na_2_O). There were three application frequencies by day intervals (DI): 0DI (without any application), 7DI (12× within 12 weeks after transplanting (WAT)), 15DI (6× within 12 WAT) and 30DI (3× within 12 WAT). From these findings, we observed that the photosynthesis rate started to increase from 10.6 to 19.4 µmol CO_2_ m^−2^s^−1^, when the total chlorophyll content started to increase from 3.85 to 7.61 mgcm^−2^. The transpiration rate started to increase from a value of 1.94 to 4.31 mmol H_2_O m^−2^s^−1^, when the stomata conductance started to increase from 0.237 to 0.958 mmol m^−2^s^−^^1^. The proline content started to increase from 22.89 to 55.07 µmg^−1^, when the relative water content started to increase from 42.92 to 83.57%.

## 1. Introduction

Banana (*Musa* spp.) is classified as non-seasonal fruit crop that belongs to the family Musaceae. The stunted growth performance of banana in farms commonly due to the impact of climate change, deficiency of mineral content in the soil, and soil-borne diseases and pests are among the constraints that have seriously and consistently affected banana production [1]. However, banana is relatively sensitive to *Fusarium* wilt disease caused by *Fusarium oxysporum* f.sp. *cubense* (FOC) in extreme climatic areas in high and lowlands [2,3]. Currently, banana plantations exist in 135 countries around the world across the tropics and subtropics in order to provide this staple food for >400 million people; however, more than 100,000 acres of banana plantations are seriously threatened around the world and cost around USD 2 billion due to the soil-borne fungus that causes the *Fusarium* wilt disease infestation [4,5,6]. The causative agent of FOC is classified into three races: FOC 1, FOC 2 and FOC 4. The tropical race 4 (FOC-TR4) has spread dramatically in the eastern hemisphere, especially in Malaysia, and has completely wiped-out banana plantations [7,8,9]. The impact of global climate change has caused banana plants to become susceptible to the disease and has challenged the banana growers’ ability to manage *Fusarium* wilt disease in subtropical or tropical areas [10]. Most plant diseases are aided by rains, high air humidity and soil moisture for the establishment and spread of disease-causing pathogens. However, moisture is a commonly recognized environmental factor that controls the transmission of fungal entomopathogens [11]. Before inducing the blockage of water uptake by roots, the pathogen of FOC will infect the xylem vessels and vascular bundles of the banana [12,13].

In general, banana plants require large amounts of fertilizer for growing well, especially nitrogen (N), phosphorus (P) and potassium (K), but due to their shallow roots, it can only obtain a small amount nutrients from a shallow depth of soil [14]. Nowadays, banana growers are confronted with high fertilizer costs and a vast amount of fertilizer being required for their plants, but the continuous application of chemical inputs alone may affect our environment and have health and safety concerns [15,16]. Therefore, the use of silicate (Si) compounds with nutrient enrichment for soil amendment is important to enhance growth performance and mitigate the biotic and abiotic stresses suffered by banana plants due to climate change, deficiency of plant nutrients and pathogens. According to [17], the application of Si compounds, such as potassium silicate (K_2_SiO_3_) and sodium silicate (Na_2_SiO_3_), potentially stimulates growth and plant defense mechanisms against pathogens for controlling various types of plant diseases. The deposition of Si compounds on the root surface may provide a source of nutrients for plants to survive under biotic or abiotic stress conditions as well as protection from FOC’s entry into the epidermal cell wall [18]. Essentially, in this study, it is important to point out the effectiveness of Si compounds with nutrient enrichment applied at different frequencies on the growth, physiological responses and biochemical changes of banana seedlings at the nursery stage.

## 2. Results

### 2.1. Crop Growth Performance

Plant height presented as the pseudo-stem height of banana was highly significantly affected (*p* < 0.05) by the treatments after transplanting. Figure 1A indicates the growth performance after 12 WAT; the average plant heights of the banana seedlings were 31.2, 28.55 and 27.85 cm for T1, T2 and T3, respectively, with T0 serving as the control uninoculated FOC (24.22 cm). Regarding the adversative effect of *Fusarium* inoculation (Figure 1B), there was a substantial reduction in control inoculated FOC equal to 38.39% at 12 WAT as compared to the control uninoculated FOC. The exogenous application of Si compounds by drenching the soil under *Fusarium* stress conditions significantly increased the height of the banana seedling by 69.23% (T1), 53.28% (T2) and 87.13% (T3) as compared to the control inoculated FOC (14.92 cm). Similarly, the pseudo-stem diameter size was significantly affected (*p* < 0.05) and Figure 2 shows the high interaction effects of the treatments on the banana seedlings and Si compounds on the pseudo-stem diameter size. There was a significant increase by 18.97%, 7.53% and 3.91% on the pseudo-stem diameter size for, T1, T2 and T3, respectively, as compared to T0 (24.22 cm) serving as the control uninoculated FOC (−ve FOC). The pseudo-stem diameter was markedly affected with *Fusarium* stress conditions, under which the reduction was 88.23% in the control inoculated FOC equal to 2.85 cm at 12 WAT as compared to the control uninoculated FOC. However, there were numerical differences between the application of Si compounds under *Fusarium* stress conditions: the pseudo-stem diameter of the Berangan seedlings significantly increased by 26.31% (T1), 30.52% (T2) and 34.03% (T3), as compared to the control inoculated FOC.

The results from Table 1 illustrate that the effect of the treatments of the banana seedlings on the root dry weight, shoot dry weight and root to shoot ratio showed no significant differences (*p* < 0.05). Indeed, the exogenous application of Si compounds on the soil planting media significantly affected (*p* < 0.05) root dry weight, shoot dry weight and root to shoot ratio. The average root dry weights of the Berangan banana seedlings after 12 WAT were T1, which had the highest (3.37 g), followed by T2 (3.6 g) and T0 (3.53 g), while T3 had the lowest root dry weight (3.5 g). The opposite occurred when the shoot dry weight of the banana seedlings treated with Si compounds significantly increased by 91.47% (T1), 92.05% (T2) and 83.11% (T3) relative to the T0 serving as the control inoculated FOC (12.08 g). The segregation between shoot and root growths can be obained based on rate of the root to shoot ratio, which depends on the dry biomass distribution to specific organs. The analysis of variance showed a significant difference (*p* < 0.05) in root to shoot ratio at 12 WAT; T0 serving as the control had the highest value at 0.226, followed by T3 (0.159) and T2 (0.156), whereas T1 had the lowest value at 0.146. Thus, the exogenous application of Si compounds on soil planting media significantly reduced the root to shoot ratio. The aerial parts of the banana increased in weight, more than the roots, even though the roots of the plant are important as they are able to supply nutrients and water from the soil to the shoots, which are the aerial parts. 

### 2.2. Total Chlorophyll Content and Leaf Gas Exchange

From the interaction effects in Figure 3, the results show that the total chlorophyll (Chl_a+b_) content of the Berangan seedlings was significantly and noticeably increased (*p* < 0.05) when Si compounds were applied by drenching the soil in the polybag. The average Chl_a+b_ contents of the Berangan banana seedlings were 7.25, 6.82 and 6.55 mgcm^−2^ for T1, T2 and T3, respectively. A reduction in the Chl_a+b_ content in the banana by 30.43% in the control inoculated FOC as compared to the control uninoculated FOC (6.44 mgcm^−2^) was observed. Moreover, the application of Si compounds under *Fusarium* stress conditions significantly increased the concentration of Chl_a+b_ content by 4.01% (T1), 18% (T2) and 18.30% (T3) as compared to T0 serving as the control inoculated FOC (4.48 mgcm^−2^). Figure 4 demonstrates the significant interaction among these factors (*p* < 0.05) on the apparent photosynthesis rate (Ps) of banana seedlings. The effect of Chl_a+b_ content significantly increased the Ps rate by 6.39% (T1), 6.79% (T2) and 5.37% (T3) when the Si compounds were applied on the banana seedlings as compared to T0 serving as the control uninoculated FOC (17.67 µmol CO_2_ m^−2^s^−1^). The Ps of the Berangan seedlings was reduced considerably with rising *Fusarium* stress conditions. The result of Ps is shown, thus indicating that the average Ps of the Berangan banana seedlings after 6 WAT was T3, which had the highest Ps with a mean value of 16.05 µmol CO_2_ m^−2^s^−1^, followed by T2 (15.9 µmol CO_2_ m^−2^s^−1^) and T1 (12.95 µmol CO_2_ m^−2^s^−1^), while T0, serving as the control inoculated FOC, had the lowest Ps, at about 12.6 µmol CO_2_ m^−2^s^−1^.

Table 2 shows that the application of Si compounds on the soil planting media of the banana seedlings had a significantly impact on stomata conductance and the transpiration rate, but there was no significant interaction effect between these factors. The stomata conductance and transpiration rate at 6 WAT were significantly and substantially reduced by 27.02% and 29.79% with rising biotic stress conditions as compared to the control uninoculated FOC. Interestingly, the banana seedlings treated with Si compounds significantly increased by 5.66% (T1) and with their corresponding values of 39.62% for T2 and T3 in comparison to the T0 serving as the control inoculated FOC (0.53 mmol m^−2^s^−1^). On the contrary, the mean comparison of the Berangan banana seedlings after 6 WAT, T2 had the highest transpiration rate (3.64 mmol H_2_O m^−2^s^−1^), followed by T3 (3.61 mmol H_2_O m^−2^s^−1^) and T1 (3.05 mmol H_2_O m^−2^s^−1^), while T0, serving as the control, had the lowest transpiration rate (3.03 mmol H_2_O m^−2^s^−1^).

### 2.3. Physiological Attributes and Biochemical Content

The relative water content (RWC) in the leaves was significantly affected under different application frequencies; moreover, obvious differences were also observed among the treatments applied on the banana seedlings. Simultaneously, there was a high significant interaction observed in RWC among the different treatments on the banana seedlings and the application frequencies of Si compounds. Regarding the adversative effects of *Fusarium* stress conditions, there was a substantial reduction in the control inoculated FOC of up to 19.87% at 12 WAT as compared to the control uninoculated FOC (67.85%). The bar chart in Figure 5 also shows the results of RWC from the banana plant treated by T1, which significantly increased by 6.79% (7DI), 6.07% (15DI) and 10.27% (30DI), as compared to T0 serving as the control uninoculated FOC (67.85%). However, the application of T3 by drenching the soil under biotic stress conditions significantly increased RWC by 17.74% (7DI), 32.09% (15DI) and 24.01% (30DI) as compared to T0 serving as the control inoculated FOC (47.98%). The bar chart in Figure 6 shows that root electrolyte leakage (EL) was significantly affected under different application frequencies, but it was more evident when the banana seedlings were exposed to *Fusarium* stress conditions. There was a significantly increased root EL of up to 10.12% for the positive control (+ve FOC), when the banana seedlings were exposed to *Fusarium* stress conditions, in comparison to the control uninoculated FOC (67.85%). The bar chart also shows the results of the root EL of the banana plant treated by T1, which was significantly reduced by 3.28% (7DI), 21.19% (15DI) and 16.38% (30DI) as compared to T0 serving as the control uninoculated FOC (64.52%). Interestingly, the application of T3 by drenching the soil under *Fusarium* stress conditions significantly decreased root EL by 22.52% (7DI), 27.31% (15DI) and 21.57% (30DI) as compared to T0 serving as the control inoculated FOC (74.64%).

Proline content was noticeably affected (*p* < 0.05) with high significant interactions among the treatments on the banana seedlings and the application frequencies of Si compounds. From the interaction results in Figure 7, we can observe that the proline content significantly increased by 39.15% in the plant tissues of banana seedlings infected with *Fusarium* wilt disease as compared to the negative control plant (−ve FOC). The exogenous application of T1 by drenching the planting media significantly reduced the accumulation of proline content in the Berangan seedlings by 25.46% (7DI), 12.74% (15DI) and 21.16% (30DI) as compared to the control uninoculated FOC (34.68 mg^−1^FW). The accumulation of proline content significantly decreased by 26.58% (7DI), 44.28% (15DI) and 39.14% (30DI) in comparison to the 0DI serving as the control inoculated FOC (48.26 mg^−1^FW), even though biotic stress rose due to *Fusarium* wilt disease.

### 2.4. Plant Nutrient Uptake

The bar chart in Figure 8 shows that the accumulation of N and P in banana leaves significantly reduced N uptake by 19.60% and P uptake by 12.30%, when the banana was infected by *Fusarium* wilt disease as compared to the control uninoculated FOC. Regarding the mean comparison on the accumulation of N in the banana leaves when T1 was applied on the banana seedlings, at 15DI was the highest concentration (1.27%), followed by 7DI (0.71%) and 30DI (0.58%), whereas 0DI, serving as the control uninoculated FOC, had the lowest N concentration was 0.51%. In contrast, the exogenous application of T3 on the banana seedlings under *Fusarium* stress conditions significantly increased the N concentration by 0.15% (7DI), 0.32% (15DI) and 0.18% (30DI) as compared to 0DI serving as the control inoculated FOC (0.41%). The application of T1 on the soil planting media without FOC significantly increased P composition in the leaves of the sampled banana plants by 0.04% (7DI), 0.01% (15DI) and 0.03% (30DI) as compared to 0DI serving as the control inoculated FOC (0.65%). The accumulation of P uptake in the banana plants treated with T3 on the infected soil with FOC significantly rose by 0.08% (7DI), followed by 0.09% (15DI) and 0.04% (30DI) as compared to 0DI serving as the control inoculated FOC (0.57%).

Table 3 shows that the content of K, Ca and Mg in the leaves of the sampled banana plants considerably had no significant interaction, but there was a significant effect (*p* < 0.05) between the application frequencies of the Si compounds. However, there were no significant effects on these three elements for both the healthy seedlings and infected seedlings (+ve FOC). The K composition of the banana leaves significantly increased by 0.23% (7DI) and 0.18% (15DI), but the accumulation of K was markedly decreased by 0.04% (30DI) in comparison to 0DI serving as the control (1.60%). In this study, the analysis of variance showed a significant difference (*p* < 0.05) on the concentration of Ca at 12 WAT; 7DI had the highest concentration at 0.41%, followed by 30DI (0.40%) and 0DI (0.38%), whereas 15DI had the lowest concentration of Ca about 0.38%. Compared to the control plants (0.46%), the accumulation of Mg in the leaves of banana plants significantly increased by 0.08% (7DI), 0.10% (15DI) and 0.06% (30DI).

### 2.5. Disease Assessment

The appearance of disease symptoms between the treatments was observed and the disease incidence (DI) measure was used to determine the effectiveness of the treatments in suppressing FOC. Disease suppression is indicated by a lower DI value. As shown in Figure 9, no DI was found at 2 WAT, but the appearance of disease symptoms in treatment T3 were observed, however disease onset was delayed until 4 WAT by the inoculation of the seedlings with FOC. The disease symptoms in the Berangan seedlings were gradually increased with a DI of 33.59% at 12 WAT of observation. Meanwhile, the yellowing of the leaves and white mycelia was observed on the planting media as well as the roots of the Berangan seedlings. These results suggest that drenching the soil planting media and roots of the banana plants with T3 produced a good level of disease suppression. Disease symptoms appeared much earlier on the Berangan seedlings without being treated by Si compounds. As expected, the untreated seedlings had the highest DI by 84.37% at 12 WAT of assessment. The application of T3 on soil planting media as well as the root of infected plants significantly reduced DI by 50.78% (7DI), 58.59% (15DI) and 31.25% (30DI) as compared to the untreated plants (0DI) at 12 WAT of assessment. However, no DI was found for the uninoculated seedlings (−ve FOC) because they did not display any *Fusarium* wilt symptoms throughout the 12 WAT of assessment.

## 3. Discussion

Based on banana growth performance and physio-biochemical changes, the outcomes of two separate greenhouse experiments were investigated. The exogenous soil application of Si compounds to the banana seedlings improved growth performance in addition to controlling *Fusarium* wilt disease at the nursery stage. According to [19], soil application with Si element results in a more efficient and effective nutrient uptake than foliar treatment. Soil application with Si compound significantly improved the nutrient absorption capacity, leaf nutrient content, yield and quality of banana plants [20], and rice plants also had an improved grain quality [21]. When the pathogen invaded the roots of the banana, the measurement of the pseudo-stem diameter, plant height increment and biomass production markedly decreased. According to [22], banana plants infected with the fungus had a lower plant height, smaller pseudo-stem diameter, wilting, and yellowing of leaves, which also dropped. Din et al. (2018) [23] also found that, when banana roots were infected with FOC, their pseudo-stem size, root size dispersion and plant height decreased. The total dry biomass indicated that the partitioning of resources were transformed in banana plants; however, the dry biomass output of the banana seedlings decreased as a result of the biotic stress caused by FOC infection, notably in the leaves and roots [24,25,26]. Thus, the enhanced Si compound fertilization applied to the diseased plants under biotic stress significantly increased the pseudo-stem size, increased plant height and boosted total dry biomass production [27].

There was a strong significant and positive relationship between Chl_a+b_ content and Ps as influenced by the treatments and application of enriched Si on the banana plants (Figure 10). The level of Ps started to increase from 10.6 to 19.4 µmol CO_2_ m^−2^s^−1^ when the Chl_a+b_ content of the banana leaf tissues started to increase from 3.85 to 7.61 mgcm^−2^. According to [28], Si compound application on the soil improved plant growth performance, increased the amount of Chl_a+b_ content and carotenoids and the total proline content. The treatment with Si compounds considerably enhanced the Chl_a+b_ content and improved the performance of the photochemical reaction linked with the Ps rate and has considerable promise for reducing the severity of numerous disorders.

Costa et al. (2021) [29] stated that K_2_SiO_3_ and Na_2_SiO_3_ applied to banana plants significantly increased chlorophyll a and b and Chl_a+b_ content in banana plant leaves. Surprisingly, the infected roots of the banana plants without any treatments for *Fusarium* wilt disease had a significant reduction in Chl_a+b_ content of up to 30.43%, and this contributed to the lower leaf gas exchange as compared to the Si-treated plants. According to [30], reduction the levels of photosynthetic pigments in the leaves were due to a decrease in the Chl_a+b_ content, which had a substantial impact on the photosynthetic system and the intake of nutrients and water to be transmitted throughout the entire tomato plant system. The results of the correlation analysis showed that the Chl_a+b_ content in the leaf tissues had a close association with the Ps of the banana plants.

The correlation analysis showed that stomata conductance and the transpiration rate, as influenced by the treatments and application of enriched Si on the banana plants, had a significant positive correlation (Figure 11). The transpiration rate started to increase from a value of 1.94 to 4.31 mmol H_2_O m^−2^s^−1^ when the stomata conductance of the banana plants started to increase from 0.237 to 0.958 mmol m^−2^s^−1^. With increased *Fusarium* and water stress conditions, the Ps rate of the bananas slows down or even stops due to a reduction in stomatal conductance [31]. Song et al. (2021) [32] found that soil treated with Si compounds absorbed by roots from the rhizospheric soil increased the photosynthetic ability of plants growing under normal conditions, but under biotic stress. According to [33], wheat cultivars treated with Si compounds had a higher Chl_a+b_ content and enhanced Ps, but the stomatal conductance decreased along with the transpiration rate when Si was applied to the roots. Reduced stomatal conductance in response to a water-deprivation-like scenario in plants, according to [34], suggests that the stomata are closed to limit water loss via the transpiration process.

As anticipated, based on the previous findings, the treatments with Si compounds were shown to be efficient growth enhancers in suppressing *Fusarium* wilt disease, which is translated in an enhanced physiological and morphological growth performance in the banana seedlings. Therefore, further investigations on the application frequency of Si compounds on soil planting were conducted to control *Fusarium* wilt disease in banana plants. As highlighted in the Results Section, Si compounds applied at different application frequencies on the roots of banana plants showed better results in terms of growth, physiology and biochemical changes to control *Fusarium* wilt disease. Most times, the Si compounds of T1 as well as T3 during the 15DI application were found to significantly increase RWC and nutrient uptake efficacy, but to reduce EL and proline content as compared to the control (0DI). The results of the correlation analysis showed that RWC is in close association with the proline content from the sampled leaf tissues of the Berangan banana plants. There was a strong significant and positive relationship between RWC and the proline content influenced by soil health conditions based on the treatments applied and the enriched Si compound application on the banana plants (Figure 12). The level of proline content started to increase from 22.89 to 55.07 µmg^−1^ when the RWC of the banana leaf tissues started to increase from 42.92 to 83.57%.

The pathogen released fusaric acid as the infection progressed, which became a key contributor in intercellular EL and cell membrane breakdown, resulting in uncontrolled leaf water loss and a lowered RWC [35,36]. The increase in EL was accompanied by an increase in cell permeability, indicating that this is a strategy for the development of disease resistance because it triggers programmed cell death while also producing reactive oxygen species to combat disease infection [35,37]. According to [38], Si compound application improves plant stress tolerance by inducing chemical endurance and strengthening cell walls to build a better mechanical barrier to inhibit pathogens from accessing the host’s tissues. When the banana seedlings were infected with FOC, proline accumulation levels significantly increased in the banana plant tissues. It was once thought to be an effective defense against oxidative stress and disease infection [39].

Si compound application led to reduced stress symptoms and significantly reduced proline content and EL in the plant tissues after disease infection [40,41]. Interestingly, the application of Si compounds either with T3 or T1 at 15DI resulted in the highest RWC and lowest EL, as compared to the other treatments. These findings show that leaf RWC and root EL always reflect the current level of metabolic activity in order to reduce the impacts of plant stress and improve plant growth [42]. A higher RWC may help the plant to combat oxidative damage in a biotic stress environment, whereas a higher EL is frequently related to an increased cell permeability and loss of integrity [43]. The usage of the Si compound as a plant booster in the soil was also found to be effective in reducing the incidence of *Fusarium* wilt disease in banana plants [44]. In line with this, applying T3 to infected banana roots at 15DI significantly reduced DI by 58.59% and DS by 58.2%, while increasing the nutrient uptake of N, P, K, Ca and Mg in the banana leaves by 0.32%, 0.09%, 0.18%, 0.38% and 0.10%, respectively. Our results indicate that the addition of Si could strengthen the plant by reducing the EL and proline content, but aiding in the uptake of minerals, especially N uptake. In the infected plants, the PS increased, P uptake was increased and water loss was reduced. As a result, the overall condition of the plants increased, which showed a higher plant growth performance. A well-nourished and hydrated plant has a better chance of resisting a fungal infection. Sealing the plant wall by embedding Si creates a physical barrier against the ingress of pathogens. In addition, the growth of *Fusarium* was limited by the pH of the soil after adding Si. Our experience of growing *Fusarium* on Si-supplemented media showed that the pH of the soil increased up to 6.7; thus, *Fusarium* grew less at a higher pH.

However, our results coincide with the findings of *Fusarium* in tropical lake soils managed with mineral fertilization, which reduced the growth rates in the *Fusarium* species complex [45,46]. Likewise, studies of tropical soils in low areas, generally depressed and flooded areas with serious drainage problems, showed that the greatest diversity of species, high biological activity and the highest population size were determined in the rhizosphere soil in Musaceae [47]. Recent studies in tropical climate for banana-growing areas established that the areas most affected by banana wilt disease have very silty soils with drainage problems, certain nutrient deficiencies and nutritional imbalances, related to the natural condition of lacustrine soils and the lack of appropriate fertilization cycles [48,49,50], which established that the colonization of roots by complexes of *Fusarium* species, especially FOC, was accompanied by plant pathologies associated with a fungus–bacteria complex due to the presence of bacteria (*Pectobacterium* and *Erwinia* genera) and fungi (*F. moniliforme*, *F. oxysporum* and *F. solani*. This suggests a more parasitic character with Musaceae, such as host plant and silicate-free soil amendments or with low concentrations. These results reveal that drenching the soil in the polybag with Si compounds has better effects on improving nutrient uptake and controlling *Fusarium* wilt disease in the banana plants. Therefore, the soil application of T1 at 15DI is recommended for use in uninoculated soils with healthy seedlings, whereas the T3 application on soil planting media at 15DI is recommended for use in infected banana seedlings to produce the best results in terms of growth performance, physio-biochemical changes and managing *Fusarium* wilt disease.

## 4. Materials and Methods

### 4.1. Experimental Materials’ Preparation and Treatment

Two separate greenhouse experiments with a similar experimental set up were conducted under a rain shelter at the University of Putra Malaysia (UPM), Selangor. The mean daily temperature was between 36 °C (day) and 26 °C (night), with corresponding average relative humidity values of 65% and 70%, respectively, and an average light intensity of 400 µmol m^−2^s^−1^ under the rain shelter. Although other environmental factors in the greenhouse offered some protection and were considered uniform throughout the experiment, the plants were exposed to the climatic conditions of the banana-growing region. Two-week-old Berangan banana seedlings were obtained from NNS Permata Holding, Malaysia. Each polybag consisted of only one healthy banana seedling. Munchong soil series with pH 4.68, having a clay texture (62.7% clay, 10.89% silt and 26.21% sand) and 1.27% organic carbon, were used as cultivation media in 15 cm × 15 cm polybags. The nutrient status of soil planting media before planting was 0.20% total N, 32 mg kg^−1^ available P and 57 mg kg^−1^ available K. This soil type is commonly used for growing bananas in Malaysia; thus, it was also used to test the plant and root growth performance of the banana seedlings at the nursery stage.

The experiment layout was a split-plot system in a randomized complete block design (RCBD) with four replications. In the first experiment, the treatments in the main plots were divided into healthy plants without FOC inoculation, which were considered as the negative control (−ve FOC), and diseased plants with FOC inoculation, which were considered as the positive control (+ve FOC) treatment. Si compounds were prepared at a constant concentration of 0.1%, but each treatment was combined with different types and proportions of plant nutrients. Each Si-based compounds was assigned as a sub-plot: T0 (control), T1 (13% SiO_2_:20% K_2_O), T2 (26.6% SiO_2_:13.4% K_2_O) and T3 (36.2% SiO_2_:17% Na_2_O). In the second experiment, T1 and T3 were selected to be drenched in the soil planting media of the healthy plants without FOC inoculation and diseased plants with FOC inoculation, respectively. The sub-sub-plots were further divided into four different application frequencies by day intervals (DI): 0DI (without any application), 7DI (12× within 12 weeks after transplanting (WAT)), 15DI (6× within 12 WAT) and 30DI (3× within 12 WAT). All agricultural practices were enforced based on the recommendations from the Malaysian Ministry of Agriculture.

Inoculum fungus FOC-TR4 suspensions were prepared from pure and fresh fungi, which were obtained from the culture collection of the Department of Plant Protection, Faculty of Agriculture, UPM. It was cultured on a potato dextrose agar medium at 27 °C in an incubator for 7 days. Discs of about 8 mm were excised from the culture. Then, suspended in distilled water and quantified using a hemocytometer. Then, the force inoculation of *Fusarium* was performed by following the protocol previously reported by [51] by dipping the banana plant roots for about 30 min in a solution of 10^6^ FOC spores per mL, with the plantlet roots treated with distilled water serving as the control. At the time of transplanting, the soil planting media around the roots of the seedlings were inoculated with 40 mL (10^8^ FOC spores mL^−1^), according to soil inoculation method described by [52]. After the soil inoculation, each seedling was transplanted into the polybag.

### 4.2. Data Collection

#### 4.2.1. Determination of Crop Growth Traits

Plant height, presented as the banana pseudo-stem height, was measured weekly throughout the experimental period, lasting 12 WAT. The plant height of the banana plants was measured from the soil surface level to the first internode located at the top of plant shoots by using a measuring tape, whereas the pseudo-stem diameter was taken by using a vernier caliper 3 cm above the soil level. At 12 WAT, the seedlings were destructively sampled and divided into roots and shoots. To remove all adhering soil particles, the roots were gently rinsed under running tap water. Before oven drying, an electronic balance was used to weigh the fresh weight of the shoots and roots. After drying in an oven at 60 °C for 72 h, the dry weight of the shoots and roots was measured with an electronic scale (Sartorious A and D FX200Iwp, Göttingen, Germany) until a constant weight was achieved. The root to shoot ratio (R:S) was calculated on a dry weight basis.

#### 4.2.2. Determination of the Total Chlorophyll Content

Chl_a+b_ was calculated using the approach described in [53]. Using a cork borer, four discs were extracted from the middle of a banana leaf and placed in a plastic vial containing 20 mL of 80% acetone (readily covered with aluminum foil). The samples were kept in the dark for 7 days to extract all of the pigments. The absorbance values of the solution from each sample were read at 647 nm and 664 nm using a spectrophotometer (UV-3101PC UV-VIS-NIR, Shimadzu, Kyoto, Japan) to determine Chl_a_, Chl_b_ and Chl_a = b_ contents, which were calculated as follows:
Chl_a_ = 13.19 (A_664_) − 2.57 (A_647_)(1)
Chl_b_ = 22.1 (A_647_) − 5.26 (A_664_)(2)
Chl_a+b_ = 3.5 (Chl_a_ + Chl_b_)/4(3)
where Chl_a_, Chl_b_ and Chl_a+b_ denote chlorophyll a, chlorophyll b and total chlorophyll (a + b), respectively. A_647_ and A_664_ represent the absorbance of the solution at 647 and 664 nm, respectively; 13.19, 2.57, 22.1 and 5.26 are the absorption coefficients; 3.5 is the total volume used in the analysis, obtained from the original solution (mL); and 4 is the total disc area (cm^2^).

#### 4.2.3. Measurement of the Leaf Gas Exchange

The leaf gas exchange was measured by using a portable photosynthesis system (Model: L1-6400, Li-COR Inc., Lincoln, NE, USA). The third fully expanded leaf was chosen from each treatment for the determination of the rate of Ps, stomata conductance and transpiration rate. The measurements used optimal conditions set at 400 µmol mol^−1^ CO_2_, 30 °C cuvette temperature and 60% relative humidity with the air flow rate set at 500 cm^3^ min^−1^. The measurement was performed at 6 WAT by clipping the leaf in the chamber at a standard time (from 0800 to 1100 am). The reading of Ps, stomata conductance and transpiration rate are expressed as µmol CO_2_ m^−2^s^−1^, mmol m^−2^s^−1^ and mmol H_2_O m^−2^s^−1^, respectively.

#### 4.2.4. Biochemical Assay of the Physiological Attributes

The RWC of the leaves was estimated according to the method of [54], while the EL of the root membrane was assessed for their membrane permeability according to [55]. Both RWC and EL are expressed in percentage (%).

The quantification of proline content in the Berangan banana leaves was performed according to the method described by [23] and expressed as µmolg^−1^ of fresh weight. The plant tissues collected from the fresh banana leaves (approximately 0.5 g) were ground in liquid nitrogen and stored for further analysis. Then, 10 mL of a 3% aqueous sulfosalicylic acid solution was added to the stored samples and filtered through Whatman paper (No. 2). A total of 2 mL of acetic acid and 2 mL of an acidic ninhydrin reagent were added to a 2 mL aliquot. The mixture was thoroughly stirred and incubated in a boiling water bath for 1 h. Subsequently, it was transferred to an ice bath and warmed to room temperature. Four milliliters (mL) were added to the mixture and the extinction of the upper toluene level was measured at 518 nm by using a spectrophotometer (Model: Shimadzu UV-160A Visible Recording Spectrophotometer, Kyoto, Japan). The amount of proline content was calculated as follows:
Proline = [(µg proline mL^−1^ × 10 mL toluene)/115.5µg µmole^−1^)]/(0.5 g sample/5)(4)
where 10 mL = volume of toluene; 115.5 µg µmole^−1^ = molecular weight of proline; 0.5 g = weight of fresh sample; and 5 = dilution factor.

#### 4.2.5. Determination of Plant Nutrient Uptake

The uptake of N, P, K, Ca and Mg nutrients was measured using the method described by [20]. To conduct the plant nutritional analysis, the fully expanded third dried-leaf samples of banana were finely ground. About 0.25 g of dry ground leaf tissue was homogenized for at least two hours with 5 mL concentrated sulfuric acid (H_2_SO_4_), before adding 2 mL 50% hydrogen peroxide (H_2_O_2_) and heating up to 285 °C in a digestion chamber for 45 min. This technique was continued until the sample’s color became clear. Finally, the solutions were completed up to 100 mL with distilled water and filtered. The N and P contents were determined using an Auto Analyzer, AA (LACHART Instruments, Milwaukee, WI, USA, Model Quikchem IC + FIA 8000 Series). The K, Ca and Mg contents were determined using an Atomic Absorption Spectrophotometer, AAS (Perkin-Elmer, 5100 PC, Waltham, MA, USA). For the determination of K by atomic absorption, serial dilutions were performed for the calibration curve; the actual linear range of K was between 0 and 10 mg K L^−1^. However, the linear working range for Ca is between 0 and 10 mg Ca L^−1^, while Mg is between 0 and 1 mg Mg L^−1^.

#### 4.2.6. Disease Assessment

The *Fusarium* wilt disease symptoms on the Berangan banana plants were monitored and recorded for yellowing or wilting leaves. The number of banana plants with visible symptoms, such as chlorosis and necrosis of leaves, over the total number of plants was measured to determine the DI for each treatment [24]. To compute the percentage of DI, the following formula was used:
DI (%) = (number of infected seedlings/total number of seedlings assessed) × 100 (5)

### 4.3. Statistical Analysis

All collected data were recorded and analyzed by using an analysis of variance (ANOVA) test by a statistical analysis system (SAS 9.4) to determine the significant difference between the treatment means. Differences between separated means were stablished using the least significant difference (LSD) test, set at a *p* < 0.05 level.

## 5. Conclusions

*Fusarium* is a serious vascular wilt disease that threatens banana plants. FOC infection significantly reduced plant height, pseudo-stem diameter, total biomass production and Chl_a+b_ content as well as physiological traits. Interestingly, the best growth performance was observed when the banana seedlings, with an application of 0.1% of T3 at 15DI, significantly reduced DI by 58.2% and proline content by 44.28%, but enhanced macro-nutrient uptake and morpho-physiological growth parameters. With a decreased DI by the exogenous application of Si compound enriched with plant nutrients, there was a strong significant and positive relationship between Chl_a+b_ content and Ps. In addition, the level of proline content was enhanced by increasing RWC, but reduced EL in response to Si compound treatments. However, the 0.1% of T1 is highly recommended to be applied as a growth enhancer at 15DI on soil planting media without FOC infection to promote the root and shoot growth of banana plants. The Si compounds applied on soil planting media at 15DI created notably better changes in the plant growth and physiology of the Berangan seedlings. In the future, field trial experiments are recommended and further studies should be conducted to determine the growth performance, yield and toxicity levels of the treated banana plants.

## Figures and Tables

**Figure 1 plants-13-00542-f001:**
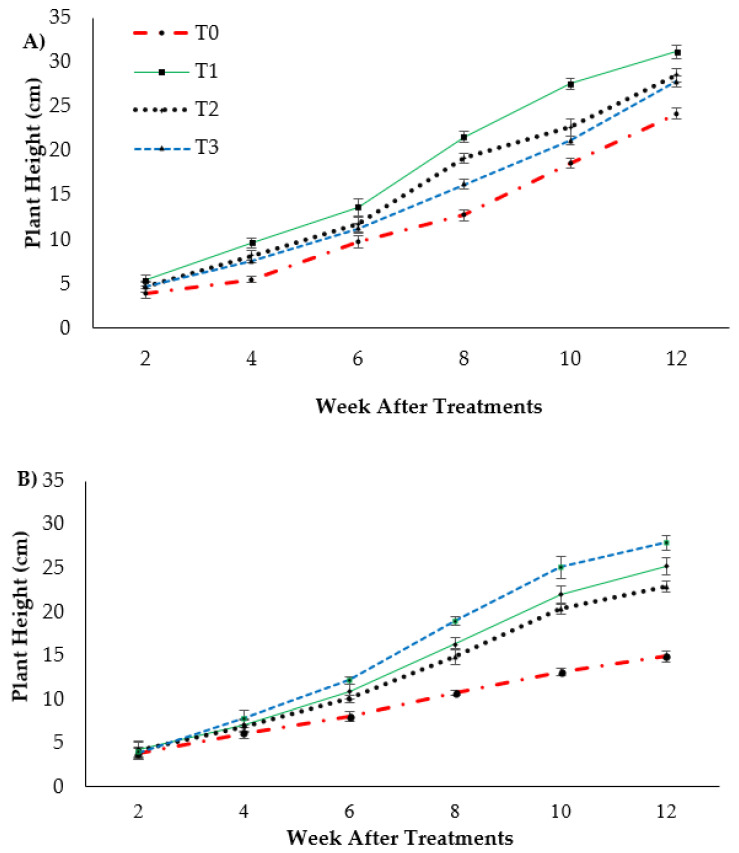
Effect of different Si compounds on changes in plant height: (**A**) banana seedlings grown on soil uninoculated FOC (−ve FOC) and (**B**) banana seedlings grown on soil inoculated FOC (+ve FOC) over a duration of 12 WAT. Data are mean ± SEM (standard error of differences between means) of 32 replicates. Soil uninoculated FOC (−ve FOC) served as the negative control and soil inoculated FOC (+ve FOC) served as the positive control. T0 = without SiO_2_ application; T1 = 13% SiO_2_ + 20% K_2_O; T2 = 26.6% SiO_2_ + 13.4% K_2_O; T3 = 36.2% SiO_2_ + 17% Na_2_O.

**Figure 2 plants-13-00542-f002:**
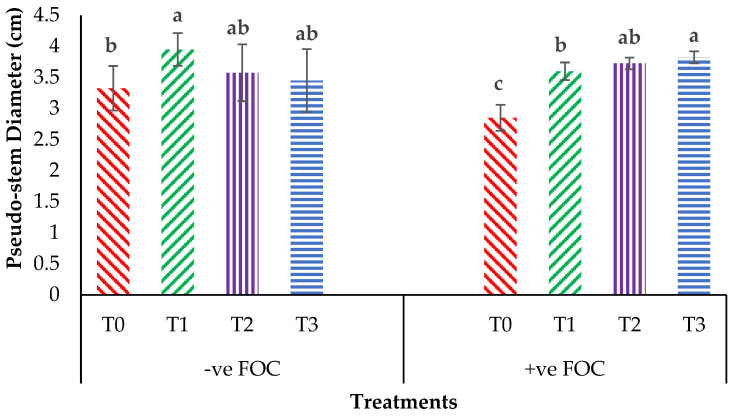
Significant interaction between the different treatments on the banana seedlings and Si compounds on the pseudo-stem diameter at 12 WAT. Data are mean ± SEM (standard error of differences between means) of 32 replicates. Bars represent means followed by the different small letters, significant at *p* < 0.05. Soil uninoculated FOC (−ve FOC) served as the negative control and soil inoculated FOC (+ve FOC) served as the positive control. T0 = without SiO_2_ application; T1 = 13% SiO_2_ + 20% K_2_O; T2 = 26.6% SiO_2_ + 13.4% K_2_O; T3 = 36.2% SiO_2_ + 17% Na_2_O.

**Figure 3 plants-13-00542-f003:**
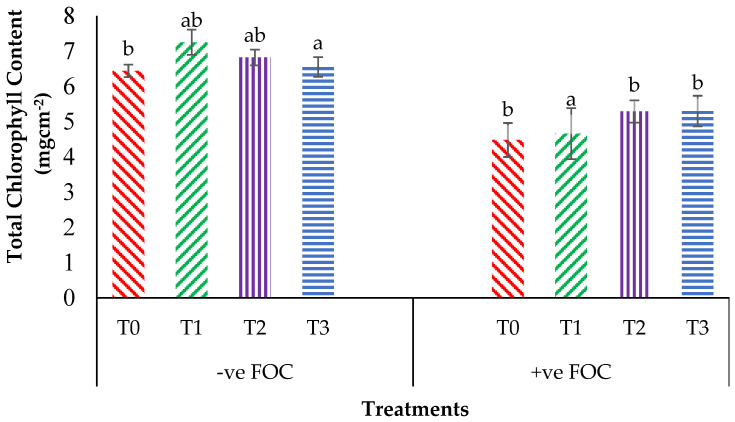
Significant interaction between the different treatments on the banana seedlings and Si compounds on the total chlorophyll content (Chl_a+b_) at 12 WAT. Data are mean ± SEM (standard error of differences between means) of 32 replicates. Bars represent means followed by the different small letters, significant at *p* < 0.05. Soil uninoculated FOC (−ve FOC) served as the negative control and soil inoculated FOC (+ve FOC) served as the positive control. T0 = without SiO_2_ application; T1 = 13% SiO_2_ + 20% K_2_O; T2 = 26.6% SiO_2_ + 13.4% K_2_O; T3 = 36.2% SiO_2_ + 17% Na_2_O.

**Figure 4 plants-13-00542-f004:**
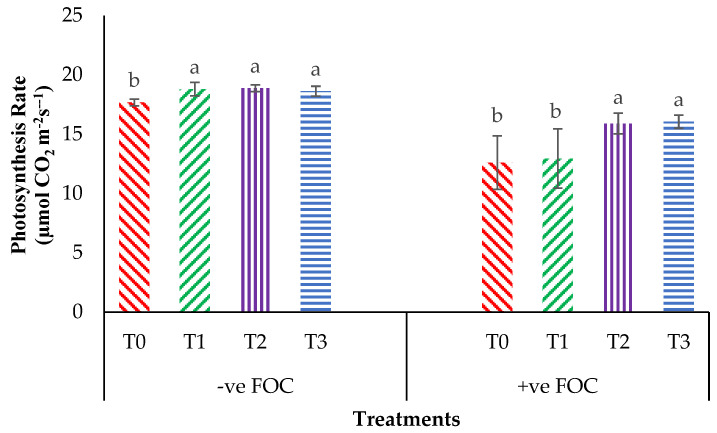
Significant interaction between the different treatments on the banana seedlings and Si compounds on the photosynthesis rate (Ps) at 6 WAT. Data are mean ± SEM (standard error of differences between means) of 32 replicates. Bars represent means followed by the different small letters, significant at *p* < 0.05. Soil uninoculated FOC (−ve FOC) served as the negative control and soil inoculated FOC (+ve FOC) served as the positive control. T0 = without SiO_2_ application; T1 = 13% SiO_2_ + 20% K_2_O; T2 = 26.6% SiO_2_ + 13.4% K_2_O; T3 = 36.2% SiO_2_ + 17% Na_2_O.

**Figure 5 plants-13-00542-f005:**
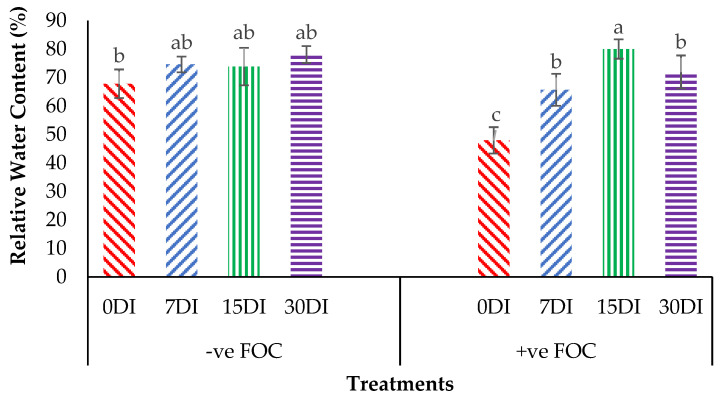
Significant interaction between the different treatments on the banana seedlings and the application frequencies of Si compounds on leaf relative water content (RWC) at 6 WAT. Data are mean ± SEM (standard error of differences between means) of 32 replicates. Bars represent means followed by the different small letters, significant at *p* < 0.05. Soil uninoculated FOC (−ve FOC) served as the negative control and soil inoculated FOC (+ve FOC) served as the positive control. 0DI = without any application; 7DI = 12 times at 7 days interval; 15DI = 6 times at 15 days interval; 30DI = 3 times at 30 days interval.

**Figure 6 plants-13-00542-f006:**
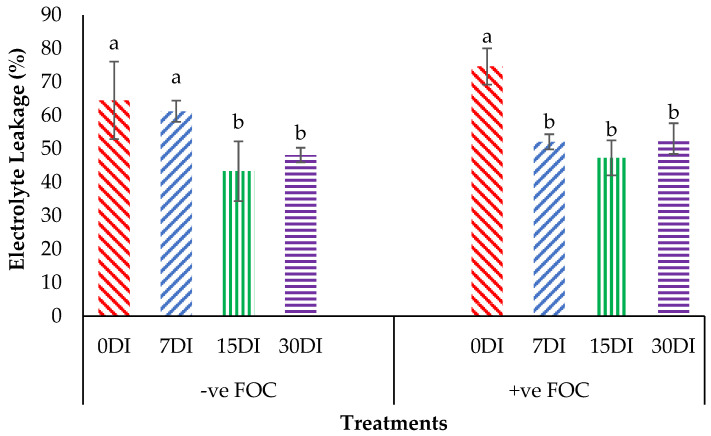
Significant interaction between the different treatments on the banana seedlings and the application frequencies of Si compounds on electrolyte leakage (EL) at 6 WAT. Data are mean ± SEM (standard error of differences between means) of 32 replicates. Bars represent means followed by the different small letters, significant at *p* < 0.05. Soil uninoculated FOC (−ve FOC) served as the negative control and soil inoculated FOC (+ve FOC) served as the positive control. 0DI = without any application; 7DI = 12 times at 7 days interval; 15DI = 6 times at 15 days interval; 30DI = 3 times at 30 days interval.

**Figure 7 plants-13-00542-f007:**
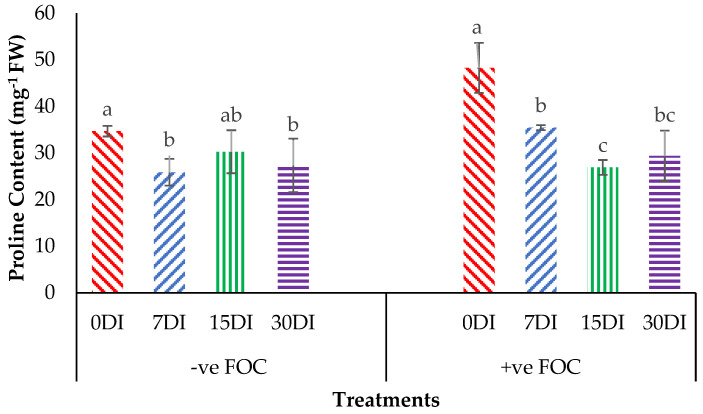
Significant interaction between the different treatments on the banana seedlings and the application frequencies of Si compounds on the proline content at 6 WAT. Data are mean ± SEM (standard error of differences between means) of 32 replicates. Bars represent means followed by the different small letters, significant at *p* < 0.05. Soil uninoculated FOC (−ve FOC) served as the negative control and soil inoculated FOC (+ve FOC) served as the positive control. 0DI = without any application; 7DI = 12 times at 7 days interval; 15DI = 6 times at 15 days interval; 30DI = 3 times at 30 days interval.

**Figure 8 plants-13-00542-f008:**
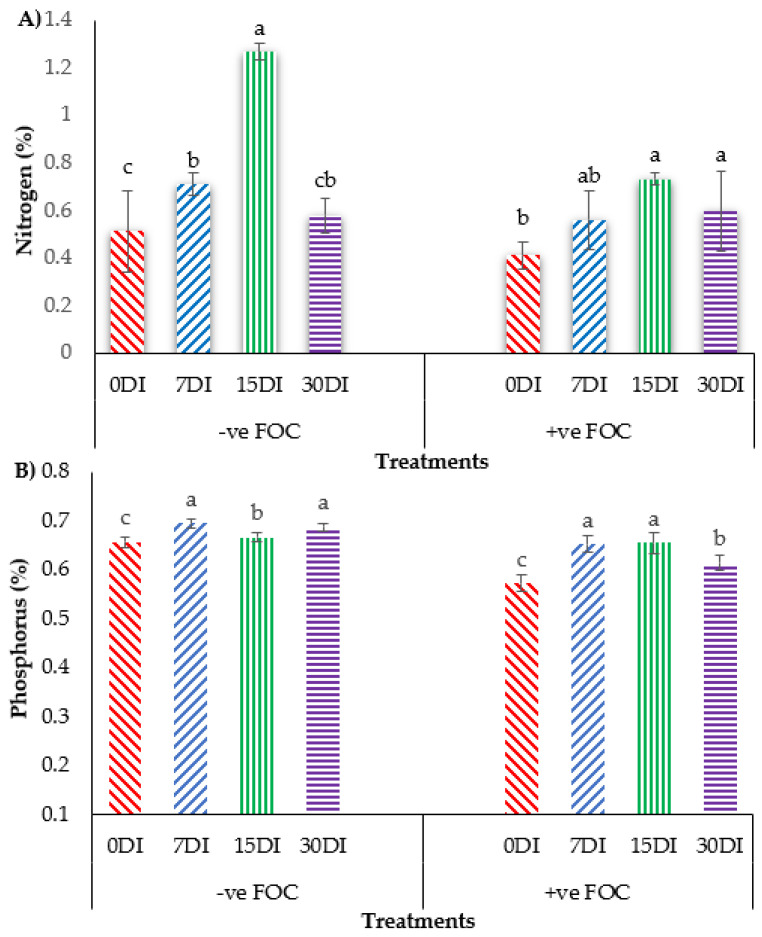
Effect of the different treatments on the banana seedlings and the application frequencies of Si compounds on macro-mineral uptake at 6 WAT: (**A**) changes in nitrogen (N) uptake and (**B**) changes in phosphorus (K) uptake. Data are mean ± SEM (standard error of differences between means) of 32 replicates. Bars represent means followed by the different small letters, significant at *p* < 0.05. Soil uninoculated FOC (−ve FOC) served as the negative control and soil inoculated FOC (+ve FOC) served as the positive control. 0DI = without any application; 7DI = 12 times at 7 days interval; 15DI = 6 times at 15 days interval; 30DI = 3 times at 30 days interval.

**Figure 9 plants-13-00542-f009:**
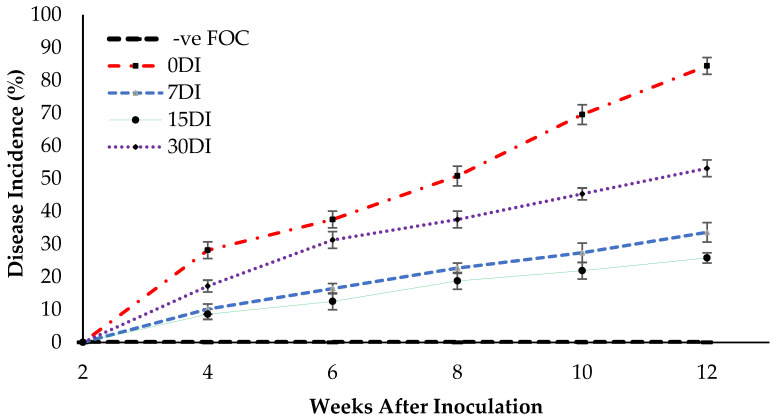
Disease incidence (DI) as affected by the different treatments on the banana seedlings and the application frequencies of Si compounds throughout the 12 WAT. Data are mean ± SEM (standard error of differences between means) by using the least significant difference test (*p* < 0.05). 0DI = without any application; 7DI = 12 times at 7 days interval; 15DI = 6 times at 15 days interval; 30DI = 3 times at 30 days interval.

**Figure 10 plants-13-00542-f010:**
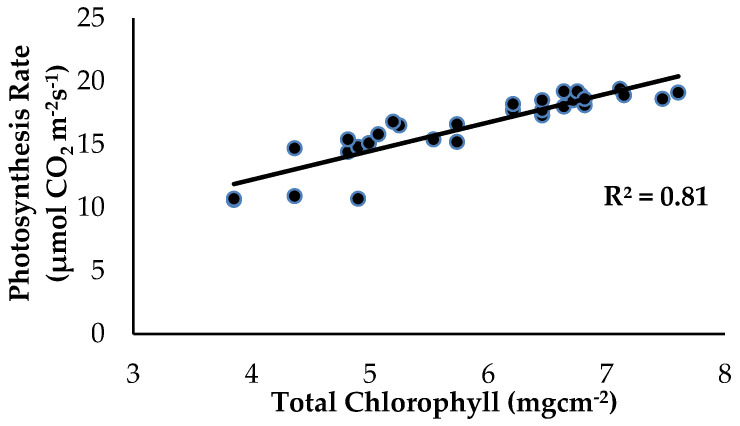
Relationship between the total chlorophyll content (Chl_a+b_) and photosynthesis rate (Ps) as influenced by the different treatments on the banana seedlings and Si compound application.

**Figure 11 plants-13-00542-f011:**
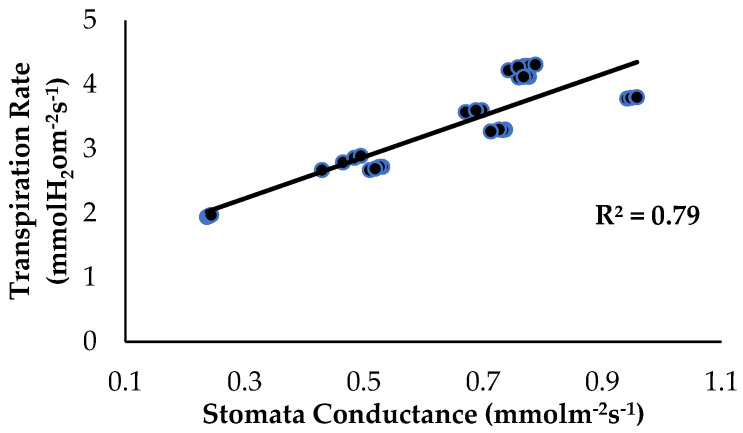
Relationship between stomata conductance and the transpiration rate as influenced by the different treatments on the banana seedlings and Si compound application.

**Figure 12 plants-13-00542-f012:**
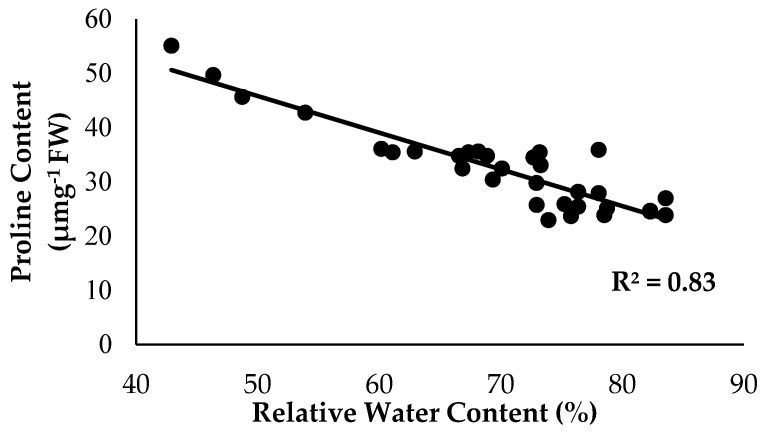
Relationship between the relative water content (RWC) and proline content as influenced by the different treatments on the banana seedlings and the application frequencies of the Si compound.

**Table 1 plants-13-00542-t001:** Effect of the treatments on the banana seedlings and Si compounds on root dry weight, shoot dry weight and root to shoot ratio at 12 WAT.

Factors	Root Dry Weight	Shoot Dry Weight	Root to Shoot Ratio
	(g)	(g)	
** Main plot means: ** **Treatments**			
−ve FOC	3.29 ± 0.45 ^a^	19.831 ± 4.55 ^a^	0.172 ± 0.03 ^a^
+ve FOC	3.21 ± 0.58 ^a^	20.444 ± 6.25 ^a^	0.173 ± 0.05 ^a^
LSD (*p* < 0.05)	NS	NS	NS
** Sub-plot means: ** **Si compounds**			
T0	2.53 ± 0.48 ^b^	12.08 ± 3.66 ^b^	0.226 ± 0.06 ^a^
T1	3.37 ± 0.2 ^a^	23.13 ± 2.21 ^a^	0.146 ± 0.01 ^b^
T2	3.6 ± 0.24 ^a^	23.2 ± 2.46 ^a^	0.156 ± 0.01 ^b^
T3	3.5 ± 0.21 ^a^	22.12 ± 2.18 ^a^	0.159 ± 0.02 ^b^
LSD (*p* < 0.05)	0.27 ***	2.84 ***	0.04 **
Significance interaction	NS	NS	NS

Means followed by the same letter within a column are not significantly different at *p* > 0.05 by the least significant difference (LSD) test with *n* = 32. ** and *** are significantly different at *p* < 0.01 and 0.001, respectively. NS = not significant. Soil uninoculated FOC (−ve FOC) served as the negative control and soil inoculated FOC (+ve FOC) served as the positive control. T0 = without SiO_2_ application; T1 = 13% SiO_2_ + 20% K_2_O; T2 = 26.6% SiO_2_ + 13.4% K_2_O; and T3 = 36.2% SiO_2_ + 17% Na_2_O.

**Table 2 plants-13-00542-t002:** Effect of the treatments on the banana seedlings and Si compounds on stomata conductance and the transpiration rate of banana plants.

Factors	Stomata Conductance	Transpiration Rate
	(mmol m^−2^s^−1^)	(mmol H_2_O m^−2^s^−1^)
** Main plot means: ** **Treatments**		
−ve FOC	0.74 ± 0.03 ^a^	3.86 ± 0.42 ^a^
+ve FOC	0.54 ± 0.26 ^b^	2.81 ±0.67 ^b^
LSD (*p* < 0.05)	0.11 *	0.39 **
** Sub-plot means: ** **Si compounds**		
T0	0.53 ±0.21 ^b^	3.03 ±0.85 ^b^
T1	0.56 ±0.22 ^b^	3.05 ± 3.05 ^b^
T2	0.74 ± 0.17 ^a^	3.64 ± 0.56 ^a^
T3	0.74 ± 0.16 ^a^	3.61 ± 0.64 ^a^
LSD (*p* < 0.05)	0.17 *	0.52 *
Significance interaction	NS	NS

Means followed by the same letter within a column are not significantly different at *p* > 0.05 by the least significant difference (LSD) with *n* = 32. * and ** are significantly different at *p* < 0.05 and 0.01, respectively, and NS = not significant. Soil uninoculated FOC (−ve FOC) served as the negative control and soil inoculated FOC (+ve FOC) served as the positive control. T0 = without SiO_2_ application; T1 = 13% SiO_2_ + 20% K_2_O; T2 = 26.6% SiO_2_ + 13.4% K_2_O; T3 = 36.2% SiO_2_ + 17% Na_2_O.

**Table 3 plants-13-00542-t003:** Effect of the different treatments on the banana seedlings and the application frequencies of Si compounds on macro-mineral content of the banana leaf tissues.

Factors	K	Ca	Mg
	(%)	(%)	(%)
** Main plot means: ** **Treatments**			
-FOC	1.74 ± 0.15 ^a^	0.39 ± 0.05 ^a^	0.53 ± 0.09 ^a^
FOC	1.65 ± 0.16 ^a^	0.38 ± 0.04 ^a^	0.52 ± 0.10 ^a^
LSD (*p* < 0.05)	NS	NS	NS
** Sub-plot means: ** **Frequencies: day interval (DI)**			
0DI	1.60 ± 0.22 ^b^	0.38 ± 0.02 ^a^	0.46 ± 0.04 ^b^
7DI	1.83 ± 0.16 ^a^	0.41 ± 0.03 ^a^	0.54 ± 0.13 ^ab^
15DI	1.78 ± 0.15 ^a^	0.33 ± 0.01 ^b^	0.56 ± 0.11 ^a^
30DI	1.56 ± 0.07 ^b^	0.40 ± 0.03 ^a^	0.52 + 0.05 ^ab^
LSD (*p* < 0.05)	0.10 ***	0.03 ***	0.10 *
Significance interaction	NS	NS	NS

Means followed by the same letter within a column are not significantly different at *p* > 0.05 by the least significant difference (LSD) test with *n* = 32. * and *** are significantly different at *p* < 0.05 and 0.001, respectively, and NS = not significant. Soil uninoculated FOC (−ve FOC) served as the negative control and soil inoculated FOC (+ve FOC) served as the positive control. 0DI = without any application; 7DI = 12 times at 7 days interval; 15DI = 6 times at 15 days interval; 30DI = 3 times at 30 days interval.

## Data Availability

Not applicable.

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
