# Peer review of "Growth Enhancement and Resistance of Banana Plants to Fusarium Wilt Disease as Affected by Silicate Compounds and Application Frequency"

_plants, 2024, doi:10.3390/plants13040542_

Round 1

Reviewer 1 Report

The manuscript, although containing some interesting results, should undergo a revision before it can be considered for publication.

My main concerns are:

1. Please enhance the introduction regarding the use of Si compounds for the soil amendment

2. In figures 1 and 9, the SEM is not very visibile. Please change the points in order to make the standard error visibile. 

3. Table 1 - Please make the statistical significance letters superscript

4. Tables 1 to 3 - the results only presents the means. Please add error

5.  Please describe the methodology, characteristics and calibration curves for the analytical determinations (especially AAS, but also the other methods)

Reviewer 2 Report

The problem of fighting banana fusariosis is still to be solved, and simple solutions are especially desirable.

Si is found in plants although it is not thought of as an element essential to the life cycle of plants.

However, plants deprived of Si are often weaker structurally and more prone to abnormalities of growth, development and reproduction

Plants differ in the accumulation of Si.

Do Authors know how Si is accumulated in banana?

It was establish that the pH allowing the uptake of Si should be more than 9.

The Authors write that the standard soil for growing bananas has pH of 4.68.

What was the pH of soil after addition of Si compounds?

Was the soli of the same pH in the control as in the samples with Si added?

After uptake Si is incorporated into the cell wall of plant and make a barrier to penetration and
enzymatic degradation by fungal pathogens.

The Authors should present how the concentration of Si in the banana cell wall increases.  This would suggest that the observed resistance to Fusarium may be related to cell wall sealing.

In addition, pH is also important for growth of Fusarium. Fungi prefer acidic pH for growth. If the pH of soil was alkaline it also could limit infections. The Authors should present how Fusarium is growing in the pH of the soil. Have the authors tried to grow Fusarium with the addition of Si supplements?

 The work describes the observations but the mechanism of action of Si added to the soil is missing. This should be added to the discussion.

In the results section the same results presented in Tables and graphical form is not necessary.  Color graphs are easier to read.

page 4

What is the conclusion from the root to shoot ratio? What is more valuable higher or lower root to shoot ratio root to shoot ratio

Methods

it is not clear how the Si compound are prepared

What are the differences among T1, T2 and T3 since Si compound has a constant concentration of 0.1%

Reviewer 3 Report

The following is a list of my comments:

Abstract

Abstract is written in a haphazard manner; writers should include their significant discoveries and link appropriately with the phrases they select to correlate.

Introduction The first piece of your article is called the introduction, and it ought to contain a distinct hypothesis and considerably build the second paragraph. Modify it so that it has a stronger connection to the statement of the issue.

Overall, there is a great deal of material that is stated many times, which need to be avoided.

Discussion

This section need to offer additional information and references connected to the works that are significant and related.

Conclusions

If it is at all feasible, the conclusion should be reorganised, thoroughly edited, and updated with clear information describing the discoveries that are most significant.

Reviewer 4 Report

This manuscript reports the growth enhancement and resistance of banana to Fusarium wilt disease as affected by silicate compounds and application frequency. The study design was fully qualitative and meets general standards and from what I can judge the data is being collected and analyzed appropriately. This work is an unprecedented effort that, to a large extent, should be made public in a scientific journal for discussion among researchers working in the field. However, it is recommended to add the suggestions, as detailed below, before it can be accepted.

Line 17: cubense on italic

Line 18: The University of Putra Malaysia (Malay: Universiti Putra Malaysia)

Line 41: cubense on italic

Line 41: FOC which has four races, with race 4 being the most aggressive (Foc, Tropical Race 4) (syn. Fusarium odoratissimum).

Line 46: should say: wiped-out

Line 394: those stomata

Line 522: the

Line 513: denotes

Line 525: am

Line 538: Four-milliliters (mL) were

Line 553: the sample’s color

Line 481: replace numbering from [45] to 51

Line 485: replace numbering from [46] to 52

Line 501: replace numbering from [47] to 53

Line 528-9: do the same with citation 48 and 49

Discussion

Line 447: I continue to add a paragraph that summarizes the importance, usefulness and social relevance, contemporary of the study, specifically pointing out the Impact, Benefit and Social Projection, something like this (for example):

Our results coincide with the findings of Fusarium in tropical lake soils managed with mineral fertilization, which reduced growth rates in the Fusarium species complex [45, 46]. Likewise, studies in tropical soils of low areas, generally depressed and flooded with serious drainage problems, showed that the greatest diversity of species, high biological activity and the highest population size was determined in the rhizosphere soil in Musaceae [47].

Recent studies in tropical banana areas establish that the most affected areas with banana wilt have very silty soils with drainage problems, certain nutrient deficiencies and nutritional imbalances, related to the natural condition of lacustrine soils and surely the lack of appropriate fertilization cycles in recent years [48,49,50], who established that the colonization of roots by complexes of Fusarium species, especially F. oxysporum, was accompanied by plant pathologies associated with a fungus-bacteria complex due to the presence of bacteria (Pectobacterium and Erwinia genera) and fungi (F. moniliforme, F. oxysporum, and F. solani, which suggests a more parasitic character with Musaceae such as host plant and silicate-free soil amendments or with low concentrations.

References

I suggest adding recent references which address the issue in question in Latin American territories. Suggested citations are for genuine scientific reasons that emphasize the current topic of study in context:

45. Campos, O.; Araya-Alman, M.; Acevedo-Opazo, C. et al. Relationship Between Soil Properties and Banana Productivity in the Two Main Cultivation Areas in Venezuela. J Soil Sci Plant Nutr 2020, 20 (3), 2512-2524.  https://doi.org/10.1007/s42729-020-00317-8

46. Olivares, B. Determination of the potential influence of soil in the differentiation of productivity and in the classification of susceptible areas to banana wilt in Venezuela. UCOPress: Spain, pp. 89-111, 2022, https://helvia.uco.es/handle/10396/22355

47. Campos, O.; Calero, J.; Rey, J.C.; Lobo, D.; Landa, B.B.; Gómez, J. A. Correlation of banana productivity levels and soil morphological properties using regularized optimal scaling regression. Catena 2022, 208, 105718. https://doi.org/10.1016/j.catena.2021.105718

48. Olivares B, Rey JC, Lobo D, Navas-Cortés JA, Gómez JA, Landa BB. Fusarium Wilt of Bananas: A Review of Agro-Environmental Factors in the Venezuelan Production System Affecting Its Development. Agronomy 2021, 11(5), 986. https://doi.org/10.3390/agronomy11050986

49. Rey, J.C; Martínez-Solórzano, G., Ramírez, H., & Pargas-Pichardo, R. Cavendish Banana Wilt, and its relationship with agroecological conditions in a lacustrine plain of Venezuela. Agronomía tropical 2020, 70, 1–12. http://doi.org/10.5281/zenodo.4346252 (In Spanish).

50. Paredes, F.; Rey, J.; Lobo, D.; Galvis-Causil, S.; Olivares, B. The relationship between the normalized difference vegetation index, rainfall, and potential evapotranspiration in a banana plantation of Venezuela. STJSSA 2021, 18(1), 58-64. http://dx.doi.org/10.20961/stjssa.v18i1.50379

Round 2

Reviewer 2 Report

The authors did not answer my questions. What was the pH of the soil after adding Si? What was the growth of Fusarium on Si supplements. This will help to pinpoint the mechanism of action of Si additions to the soil. What was the concentration of Si in the plants. Each plant takes up Si differently. Plant can embed Si into the wall and this gives a protective effect against Fusarium infection. The authors did not add anything to the Discussion that might indicate the mechanism of Si action in the case of the banana tree. Is the plant wall reinforced? Does Fusarium grow worse because the soil pH is alkaline and therefore it attacks plants less? If the pH is basic, Si may be less uptake by the plant.

In the fragment added to the discussion, the authors indicate problems in tropical lake soils with drainage and the lack of nutrients in soils and the lack of silicate.

Look below

Something like this will explain what you actually achieved in your experiences

The question is, how does Si reduce disease incidents. How does Si added to soil work? Our results indicate that the addition of Si can strengthen the plant by reducing electrolyte leakage and aiding uptake of minerals (nitrogen). In infected plants, the photosynthesis rate increases, phosphorus uptake is increased, and water loss is reduced. As a result, the overall condition of the plants increases, which shows more intensive plant growth. A well-nourished and hydrated plant has a better chance of resisting a fungal infection. Sealing the plant wall by embedding Si creates a physical barrier against the ingress of pathogens. In addition, the growth of Fusarium may be limited by the pH of the soil after adding Si. Our experience of growing Fusarium on Si supplemented media has shown .... ??? that Fusarium grows less at higher pH.

line 606-607       Nutrient status of soil was 0.20% total N, 32 mg kg-1 available P, and 57 mg kg-1 available K

line 616     different types and proportion of plant nutrient  -Do you mean K2O and Na2O

Reviewer 3 Report

The authors have significantly improved the manuscript therefore it can be accepted for publication.

Round 3

Reviewer 2 Report

The problem of banana plant infections caused by Fusarium is very difficult to solve. The search for new, cheap methods of reducing infection is very much needed.

The pH of 6.7 is not high enough to harm Fusarium. I think the mechanism for limiting infection lies in sealing the cell wall. The authors showed that electrolyte leakage is reduced during soil supplementation with Si. This indirectly testifies to the sealing of the plant cell wall. It is a pity that there is no direct evidence of an increase in the amount of Si in the wall.

To check how the addition of Si works on Fusarium, I meant Fusarium breeding on medium with the addition of Si. To be sure which factor works to reduce infection, the entire complex system of plant, soil, and Fusarium must be broken down into individual factors.

Figure1

Please use the same marks for figures A and B

for example - The green line is use for T1 in Fig1A and for T3 in Fig1B

a quick look at both figures gives the impression that nothing has changed

Figure 8 A

nitrogen uptake

The bars have borders only in this figure. All figures must be of the same design

Figure 8 A and B have frames and others do not
